**Data Availability Statement:** All data files are available on Zenodo (10.5281/zenodo.10892686) and the code is available on GitHub (https://github.

# Coral growth along a natural gradient of seawater temperature, pH, and oxygen in a nearshore seagrass bed on Dongsha Atoll, Taiwan

**Ariel K. Pezner**[1]*, **Travis A. Courtney**[1,2], **Wen-Chen Chou**[3,4], **Hui-Chuan Chu**[3], **Benjamin W. Frable**[1], **Samuel A. H. Kekuewa**[1], **Keryea Soong**[5], **Yi Wei**[5], **Andreas J. Andersson**[1]

1 Scripps Institution of Oceanography, University of California San Diego, San Diego, CA, United States of America, 2 Department of Marine Sciences, University of Puerto Rico Mayagüez, Mayagüez, Puerto Rico, 3 Institute of Marine Environment and Ecology, National Taiwan Ocean University, Keelung, Taiwan, 4 Center of Excellence for the Oceans, National Taiwan Ocean University, Keelung, Taiwan, 5 Department of Oceanography, National Sun Yat-sen University, Kaohsiung, Taiwan

* arielpezner@gmail.com

## Abstract

Coral reefs are facing threats from a variety of global change stressors, including ocean warming, acidification, and deoxygenation. It has been hypothesized that growing corals near primary producers such as macroalgae or seagrass may help to ameliorate acidification and deoxygenation stress, however few studies have explored this effect in situ. Here, we investigated differences in coral growth rates across a natural gradient in seawater temperature, pH, and dissolved oxygen (DO) variability in a nearshore seagrass bed on Dongsha Atoll, Taiwan, South China Sea. We observed strong spatial gradients in temperature (5°C), pH (0.29 pH units), and DO (129 μmol $O_2$ $kg^{-1}$) across the 1-kilometer wide seagrass bed. Similarly, diel variability recorded by an autonomous sensor in the shallow seagrass measured diel ranges in temperature, pH, and DO of up to 2.6°C, 0.55, and 204 μmol $O_2$ $kg^{-1}$, respectively. Skeletal cores collected from 15 massive *Porites* corals growing in the seagrass bed at 4 sites revealed no significant differences in coral calcification rates between sites along the gradients. However, significant differences in skeletal extension rate and density suggest that the dynamic temperature, pH, and/or DO variability may have influenced these properties. The lack of differences in coral growth between sites may be because favorable calcification conditions during the day (high temperature, pH, and DO) were proportionally balanced by unfavorable conditions during the night (low temperature, pH, and DO). Alternatively, other factors were simply more important in controlling coral calcification and/or corals were acclimated to the prevailing conditions at each site.

## Introduction

Coral reefs are experiencing a myriad of environmental changes, from warming to acidification and deoxygenation [e.g., 1–4]. At the same time, coral cover has declined worldwide in

**Funding:** This research was supported by the National Science Foundation (https://www.nsf.gov/) grants OCE-1255042 (AJA) and OCE-1829778 (AJA), a National Science Foundation Graduate Research Fellowship DGE-2038238 (AKP), and Philanthropic Educational Organization (P.E.O.; https://www.peointernational.org/peo-scholar-awards) Scholar Award (AKP). These funders did not play any role in study design, data collection and analysis, decision to publish, or preparation of the manuscript. There was no additional external funding received for this study.

**Competing interests:** The authors have declared that no competing interests exist.

recent decades, with a global loss of 13.5% between 2009 and 2020 [5]. Warming in coastal areas has led to mass coral bleaching events and significant coral mortality in tropical reefs around the world [6]. In addition, ocean acidification (OA), caused by the uptake of excess anthropogenic $CO_2$ in seawater [7], has been predicted to decrease calcification rates in scleractinian corals [8]. While the effects of both temperature and acidification on coral growth have become relatively well-studied, less is known about the impacts of low oxygen or 'deoxygenation' on coral growth [for a review, see 3, 4]. However, experimental and field evidence suggests that low oxygen conditions are prevalent on tropical coral reefs today [4] and acute, severe hypoxic events can lead to mass mortality for vulnerable species [e.g., 9, 10]. Importantly, rising temperatures, low pH, and low oxygen are not occurring in isolation, and many of the processes that cause warming, acidification, or deoxygenation are interconnected [11]. Moreover, continued warming will increase the duration, intensity, and severity of low oxygen events on global coral reefs [4].

Management strategies proposed to help ameliorate the effects of OA and deoxygenation have included proposals to grow marine photosynthesizers near corals or place corals in areas with a high cover of primary producers [12–14]. Experimental evidence suggests that primary producers like macroalgae and seagrasses can modify local water chemistry in ways to oppose OA, potentially leading to benefits for calcifiers downstream [12, 15]. Marine photosynthesizers are also predicted to potentially benefit from increased ocean acidity, as an increase in the $CO_2$ available in seawater can enhance photosynthetic rates [16, 17]. Thus, the potential refugia created by the primary producers could be maintained under increasing acidification.

Seagrasses are known to be among the most productive marine primary producers, and it is well-documented that they have the capacity to change the chemistry of the surrounding seawater [12, 18–21] over large spatial scales [22–24] and throughout the water column [25]. Daytime photosynthesis by seagrass removes dissolved inorganic carbon (DIC) from the water and produces oxygen, causing an increase in pH and dissolved oxygen (DO) concentrations [19], leading many to hypothesize that seagrasses can create a pH refuge for other species from OA [12–14, 21–22, 26–28]. Studies in some temperate and tropical calcifying marine species have shown that the presence of primary producers can relieve pH stress from acidification, leading to higher calcification rates [20, 29–32]. However, the extent of the positive effect on calcification depends on factors such as the ratio of photosynthesizer biomass to water volume [29]. In addition, models estimate that the ability of seagrasses to ameliorate OA impacts will likely vary as a function of depth-averaged and seasonal productivity as well as seawater residence time [27, 28].

While elevations of pH and DO during the daytime in areas with a high cover of primary producers may be beneficial, nighttime respiration significantly lowers pH and DO, often to or past the point of environmental hypoxia [4, 24, 33], which has typically been defined as oxygen concentrations below 2 mg $O_2$ $L^{-1}$ or ~61 µmol $O_2$ $kg^{-1}$ [34, 35]. Compared to coral reef communities, the magnitude of change in pH and DO over a single day in ecosystems dominated by primary producers is immense [4, 33, 36, 37], with extremely low pH and DO in the early morning and high pH and DO in the late afternoon. Currently, it is unclear whether the benefits of elevated pH during the day outweigh the potential costs of extremely low pH at night for calcifying organisms like corals [38, 39]. While many studies on the effects of low pH alone have found the impact to be minimal to moderate for marine calcifiers [40, 41], the effects of low and variable oxygen have been shown to have strong impacts for some species [34]. In the field, research in both seagrass and mangrove ecosystems has shown that corals can successfully grow in dynamic oxygen and pH environments, but with lower calcification and photosynthetic rates than corals found on reefs [33, 37, 39], suggesting there may be physiological tradeoffs to living in these highly variable environments.

In this study, we used a seagrass bed on Dongsha Atoll, Taiwan as a natural laboratory to investigate: 1) What is the natural spatiotemporal variability in seawater temperature, carbonate chemistry, and dissolved oxygen across the seagrass bed? and 2) How do coral growth parameters such as extension, density, and calcification rates vary across the seagrass bed? We hypothesized that coral calcification rates would be highest at the sites with the highest daily maximum temperature, pH, and DO due to enhanced metabolism and chemical conditions for mineral deposition. Alternatively, we would expect these sites to have the lowest calcification rates due to negative influence of low pH and/or hypoxia occurring during nighttime that outweigh the positive influence of daytime conditions. To address these questions and hypotheses, we characterized changes in temperature, seawater carbonate chemistry, and DO across the seagrass bed over space and time using in situ discrete sampling and autonomous observations. We also quantified annual skeletal extension, density, and calcification rates in massive *Porites* collected from four sites across the seagrass bed to assess how the chemical gradients might have influenced these growth parameters.

## Materials and methods

### Study site

Dongsha Atoll (20˚40' N, 116˚48' E) is a nearly circular coral atoll located in the northern South China Sea ~ 420 km southwest of Kaohsiung, Taiwan. Dongsha has an approximate diameter of 23 km and an area of 460 km² [42] (Fig 1). The atoll is composed of a ~ 3 km wide shallow reef flat and an inner lagoon dominated by patch reefs with an average depth of 10 m [43]. Water exchange with the open ocean occurs as water flows over the reef flat into the atoll

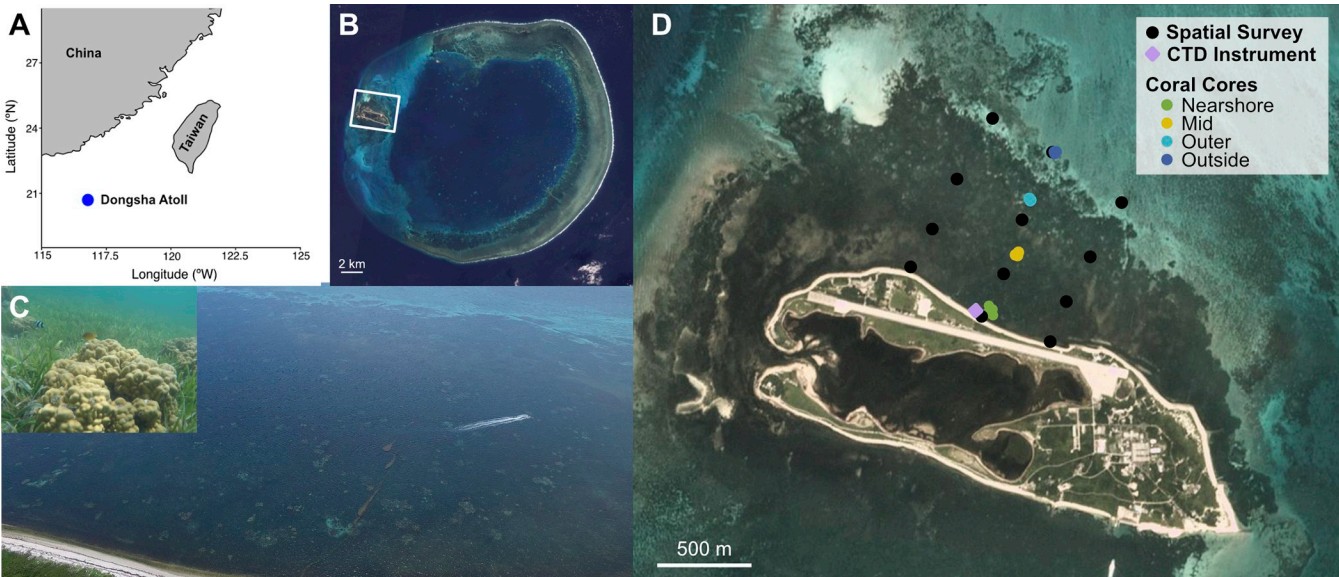

**Fig 1. Map and images of study site and sampling locations.** (A) Map of the South China Sea including coastal China, Taiwan (Republic of China), and Dongsha Atoll (blue circle). (B) Satellite image of Dongsha Atoll (Image credit: Planet Labs PBC) and Dongsha Island study area (white rectangle). (C) Drone image of the seagrass bed from above showing the extent of the seagrass from the shore to ~1 km offshore, coral colonies scattered throughout the seagrass, an inset photo of a *Porites sp.* colony in the seagrass (top left), and the progress of a jet ski moving across the seagrass (right). (D) Satellite image of Dongsha Island and surrounding seagrass beds (Image credit: Planet Labs PBC). Spatial survey locations are denoted by black circles, the CTD instrument deployment location is denoted by a purple diamond, and the coral core collection locations are shown in circles colored by collection location (nearshore–green, mid–yellow, outer–light blue, and outside–dark blue).

as well as through large open channels on the western side of the atoll both north and south of Dongsha Island, depending on the tide [44, 45].

Positioned on the western edge of the atoll, Dongsha Island (2.86 km long, 1.75 km$^2$) is home to expansive seagrass meadows that extend up to ~1 km offshore, covering an estimated area of between 8.2 and 11.85 km$^2$, depending on the survey year and season [46–48] (Fig 1C and 1D). The seagrass meadows on Dongsha have been found to contain seven species of seagrass from six genera and two families, though the meadows on the north shore where the present study was conducted are dominated by *Thalassia hemprichii*, *Cymodocea rotundata*, and *Cymodocea serrulata* [47–49]. Seagrass percent cover along the north shore was estimated to be 81% in 2011, with a mean shoot density of between 455 and 2522 shoots m$^{-2}$ [48]. The seagrass beds are shallow, with a depth of 0.5 to 2.2 m nearshore (tidal range of 0.85 ± 0.21 m; [49, this study]) and 3 m offshore [43]. Within these seagrass meadows are also some small patches of scleractinian corals, mainly massive *Porites* and some *Acropora* colonies (Fig 1C). The presence of coral colonies in the seagrass bed have been noted in the literature since at least 1994 [46–50].

**Inclusivity in global research.** Permission and consent for this research on Dongsha Atoll was provided by the director of the Dongsha Atoll Research Station at National Sun Yat-sen Univeristy, the Republic of China (Taiwan) Coast Guard, and the Dongsha Atoll Marine National Park. Permits for collection and shipping of the coral cores were approved by the Dongsha Atoll Marine National Park and CITES. Additional information regarding the ethical, cultural, and scientific considerations specific to inclusivity in global research is included in the Supporting Information (S1 File).

## Spatial surveys

To assess the spatial and temporal variability of water chemistry parameters and dissolved oxygen (DO) across the seagrass bed, seawater samples were collected at 12 stations on the north shore of Dongsha Island in 2018 (Fig 1D). The sampling area covered approximately 0.7 km$^2$ across the seagrass bed. The spatial surveys were conducted a total of four times, with one in the early morning (~6:30–7:15 h) on June 30, one mid-morning (~11:00–12:00 h) on June 27, one mid-day (~12:45–13:30 h) on June 30, and one in the late afternoon (~15:15–16:30 h) on June 26.

Spatial surveys were conducted using a jet ski, with one driver and one collector, as the water depth was too shallow for boat use. Water samples (N = 12 stations × 4 surveys) were collected at the surface in 250 mL Pyrex glass sample bottles by rinsing the bottle with seawater three times and then submerging the bottle in the water until filled (ensuring no bubbles were entrapped) at all stations. In situ temperature (± 0.2˚C), salinity (± 1%), and dissolved oxygen (± 0.2 mg L$^{-1}$) were measured with a YSI Professional Plus handheld multiparameter instrument. Once all of the samples were collected, the bottles were then handed off to scientists on the beach and immediately fixed with 100 μL of a saturated HgCl$_2$ solution and sealed for later analysis of dissolved inorganic carbon (DIC) and total alkalinity (TA), according to best practices [51].

The water samples were transported back to Scripps Institution of Oceanography (SIO), where DIC was measured using an Automated Infra-Red Inorganic Carbon Analyzer (AIRICA, Marianda) with a Li-COR 7000 detector. TA was analyzed via an open-cell potentiometric acid titration system developed by the Dickson Lab at SIO. Accuracy and precision of TA and DIC measurements were calculated as the mean offsets (± SD) from Certified Reference Materials provided by the Dickson Lab and were 1.5 ± 2.5 μmol kg$^{-1}$ (N = 23) for DIC and -0.62 ± 2.1 μmol kg$^{-1}$ (N = 20) for TA. Additional carbonate chemistry parameters were

calculated using CO2SYS for Excel [52] and the Seacarb package in R [53]. Constants and scales used included the first and second dissociation constants of carbonic acid (K1 and K2) from [54] refit by [55], the dissociation constants of bisulfate ($K_{HSO4}$) from [56] and total boron by [57], and total scale pH ($pH_T$). For TA:DIC analyses, TA and DIC were normalized to a mean salinity across all surveys of 34.2. The TA and DIC of the freshwater endmember were both assumed to be zero. Normalized DIC derived from Net Community Production ($nDIC_{NCP}$) was approximated by removing the influence of $CaCO_3$ calcification and dissolution on DIC using Eq (1):

$$nDIC_{NCP} = nDIC + \left( \frac{nTA_{mean} - nTA}{2} \right) \tag{1}$$

This calculation assumes that the mean nTA across all surveys serves as a reasonable source water value. Deviations from this assumption lead to offsets in the approximated $nDIC_{NCP}$ from its true value, which however are small in systems where organic carbon cycling is greater than $CaCO_3$ cycling, as is the case here. Furthermore, the calculation does not affect the absolute change in $nDIC_{NCP}$ between different observations, and therefore, can be accurately used to assess DO and $nDIC_{NCP}$ stoichiometry.

## Autonomous measurements

An Ocean Seven 316 Plus CTD (IDRONAUT S.r.l, Italy) was deployed at a shallow, nearshore site in the seagrass meadow to collect high frequency, temporal measurements of seawater chemistry. This deployment site overlapped with the central nearshore survey station (Fig 1D). The CTD recorded temperature (accuracy: $\pm 0.002°C$), salinity ($\pm 0.003$ PSU), DO ($\pm 0.1$ mg $L^{-1}$), and $pH_{NBS}$ ($\pm 0.01$) every 15 minutes. The CTD was deployed from 25 June 2018 until 8 July 2018. However, a problem with the battery led to a loss of data between 27 June and 1 July 2018. Thus, only data from 2 July to 8 July 2018 are reported in the present study. All CTD pH data were converted from NBS scale to total scale using the offset from the pH of a bottle sample collected next to the CTD during deployment. The pH from the bottle sample (in total scale) was calculated in CO2SYS using measured TA and DIC. The offset (-0.059) was calculated as the difference between the bottle pH and the average of the two CTD pH values at time points before and after the water sample was collected. All pH data presented here forward are reported in total scale ($pH_T$).

## Coral core collection and analysis

Coral cores were drilled from 20 individual colonies of massive *Porites* along a transect perpendicular to the north shore of Dongsha Island and along the inshore-offshore axis of the spatial surveys (Fig 1D). Cores were collected under the Dongsha Atoll National Park Permit 營海保字第1101008723號. To compare growth rates between corals across and outside of the seagrass bed, we selected 4 sites to collect cores (n = 5 per site) located: ~15 m offshore in dense seagrass ("nearshore"), ~300 m offshore in seagrass ("mid"), ~600 m offshore in a patch with no seagrass ("outer"), and ~1 km offshore outside the seagrass bed ("outside") (Fig 1D). We note that there was a depth gradient from the nearshore to outside locations, with the shallowest coral collected at ~ 0.6 m depth and the deepest coral collected at ~ 2.4 m depth (as measured by a SUUNTO Zoop Wrist Scuba Diving Computer, relative to the mean lower low water depth based on available tide data: https://dongsha-mr.nsysu.edu.tw/p/412-1272-8736. php?Lang=zh-tw), which may have impacted light intensity across the gradient. In the absence of in situ light measurements, we can estimate the difference in light intensity across this

gradient using the equation for light intensity:

$$I(z) = I(0)e^{-kz} \tag{2}$$

where $I$ is light intensity, $k$ is the light attenuation coefficient, and $z$ is depth. Assuming a constant $k$, this would result in an estimated 10% decrease in light intensity from 0.6 m to 2.4 m depth.

Cores were drilled using a handheld pneumatic drill (Nemo Power Tools) and a 3 cm drill bit from freestanding massive *Porites* colonies of approximately the same size (~50 cm tall). Holes in the corals from coring were sealed with epoxy. Cores measured 3 cm in diameter, and most were ~20 cm in length (excluding those that broke or were interrupted by low drill battery). Cores were cleaned, dried, and exported to SIO for processing under CITES Permit Number FTS507W0174606. Cores were cut using a double-bladed lapidary saw to produce an 8 mm thick cross section from the center of each core that was then x-rayed with matching aluminum and aragonite (*Tridacna sp.* shell) wedges at 35 kVp and 0.3 mAs for 40 seconds using a Faxitron X-Ray in the SIO Marine Vertebrate Collection. Scans of each core were then analyzed in Coral XDS [58] to identify growth bands and calculate annual linear extension, density, and calcification rates. Ultimately, 5 cores were excluded from the analyses (one core from the nearshore, mid, and outer sites and two cores from the outside site) due to limited or unidentifiable annual banding patterns.

## Calculations and statistical analyses

Spatial gradients in temperature, carbonate chemistry, and DO across the seagrass were calculated as the difference between the survey mean and the station value for each parameter. This was performed for each survey independently. A positive value thus indicates that the value recorded at the station was higher than the survey mean, whereas a negative value indicates a value lower than the survey mean (S1 Fig).

To statistically compare the spatial gradients within each survey, an analysis of variance (ANOVA) was performed with survey and location as fixed effects (parameter ~ Location × Survey) for temperature, DO, pH, DIC, and TA. The stations were grouped into four groups by their distance to shore (nearshore, mid, outer, and outside). The 'emmeans' package and emmeans() function with a Tukey adjustment [59] in RStudio [60] was used to compute pairwise comparisons of the locations for each survey and parameter. All tested parameters met the required assumptions of normality and homogeneity of variances required for the ANOVAs.

Similarly, to compare annual extension rates, density, and calcification rates between coral cores at different locations, linear mixed effects models were constructed with location as a fixed effect and random slopes and intercepts for each core (i.e., growth parameter ~ location, random = ~ location | core) using the lme() function from the 'nlme' package [61] in RStudio [60]. The 'emmeans' package and emmeans() function with a Tukey adjustment [59] in RStudio [60] was used to compute pairwise comparisons of these growth parameters between each location. Linear models were also created for calcification ~ density, density ~ extension, and extension ~ calcification using the lmodel2() function [62] in RStudio [60] to compute $R^2$, slopes, and p-values. Data used in this study are publicly available for download on Zenodo (10.5281/zenodo.10892686) and all code is publicly available on GitHub (https://github.com/apezner/DongshaSeagrassCores).

# Results

## Spatial variability of carbonate chemistry and dissolved oxygen

Repeat spatial surveys over the seagrass bed revealed gradients in seawater temperature, carbonate chemistry, and dissolved oxygen (DO) from nearshore to outside stations (Figs 2 and 3, S1 Fig and S1 Table). These gradients were most pronounced during the early morning and late afternoon surveys (Figs 2 and 3, S1 Fig and S1 Table).

For the early morning survey, post-hoc Tukey tests of the ANOVA of the spatial gradients for each parameter revealed that both mean DO and pH were significantly ($p < 0.01$) higher at both the outer and outside stations compared to the nearshore stations (Figs 2 and 3). DO recorded at the nearshore stations was up to 69 µmol $O_2$ $kg^{-1}$ lower than the early morning survey average and up to 129 µmol $O_2$ $kg^{-1}$ lower than the highest value recorded at the outside stations (Fig 2, S1 Fig and S1 Table). Notably, DO concentrations recorded at the three nearshore stations were at or approaching severely hypoxic concentrations (defined as $< 61$ µmol $O_2$ $kg^{-1}$; [35]) at 61, 64, and 77 µmol $O_2$ $kg^{-1}$. pH at the nearshore stations was up to 0.13 units lower than the survey average and up to 0.24 units lower than the highest pH recorded at the outside stations (Fig 2, S1 Fig and S1 Table). There were no statistically significant differences in mean temperature, DIC, and TA between the nearshore and outside stations ($p > 0.05$, Tukey test; Fig 3).

Even more pronounced spatial gradients were observed during the late afternoon survey for all measured parameters, except TA (Figs 2 and 3 and S1 Fig). Post-hoc Tukey tests of the ANOVAs of the spatial gradients for the afternoon survey revealed that mean temperature and pH at the mid, outer, and outside stations were significantly lower than the temperature and pH at the nearshore stations ($p < 0.01$), but not significantly different from each other (Fig 3). Mean DO at the nearshore stations was also significantly higher than the outside stations ($p < 0.05$), but not statistically different from the other stations (Fig 3). Temperature at the nearshore stations was up to 2.6°C warmer than the survey average and up to 5.0°C warmer than the coolest outside station (Fig 2, S1 Fig and S1 Table). DO and pH were up to 62 µmol $O_2$ $kg^{-1}$ and 0.17 units higher at the nearshore stations than the survey average, and up to 129 µmol $O_2$ $kg^{-1}$ and 0.29 units higher than the lowest values recorded at the outside stations, respectively (Fig 2, S1 Fig and S1 Table). Similarly, mean DIC at the mid, outer, and outside stations was significantly higher than the DIC at the nearshore stations ($p < 0.01$) but was not significantly different from each other (Fig 3). DIC was up to 204 µmol $kg^{-1}$ lower at the nearshore stations compared to the survey average and up to 305 µmol $kg^{-1}$ lower than the highest value recorded at the outside stations (Fig 2, S1 Fig and S1 Table). The gradients in total alkalinity, similar to the early morning survey, were less clear because both the highest and lowest TA were measured at adjacent nearshore stations (Fig 2).

Comparing observations at each station between the late afternoon and early morning surveys revealed strong temporal variability in all parameters along this spatial gradient. At the nearshore stations, temperature was up to 6.5°C warmer, the DO concentration 332 µmol $O_2$ $kg^{-1}$ higher, and pH 0.75 units higher in the late afternoon survey compared to the early morning survey (Fig 2 and S1 Table). At the outside stations, temperature was up to 3.8°C warmer, DO was 169 µmol $O_2$ $kg^{-1}$ higher, and pH was 0.40 units higher in the late afternoon compared to the early morning survey (Fig 2 and S1 Table). The differences observed between the mid and outer stations between the early morning and late afternoon surveys were of similar magnitude. In addition, the differences between all station locations between the mid-morning and mid-day surveys were much smaller than the differences observed between the early morning and late afternoon surveys.

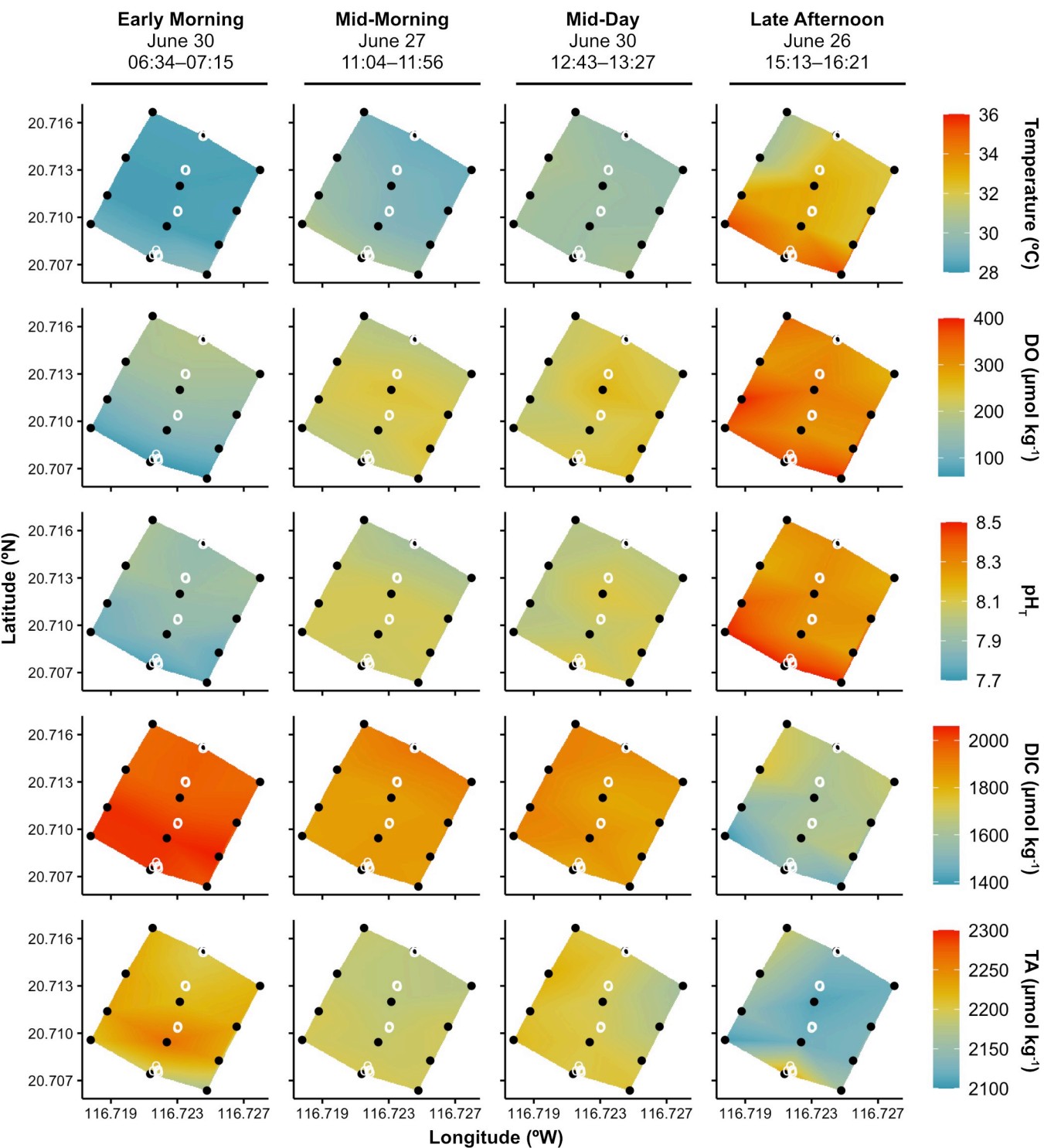

**Fig 2. Spatial surveys of seawater temperature, carbonate chemistry, and dissolved oxygen across the seagrass bed.** Spatiotemporal variability in temperature (˚C), dissolved oxygen (DO; μmol kg$^{-1}$), total scale pH (pH$_T$), dissolved inorganic carbon (DIC; μmol kg$^{-1}$); and total alkalinity (TA; μmol kg$^{-1}$) across the four spatial surveys (stations denoted by black circles) in the Dongsha Island north shore seagrass bed. Coral core collection locations are denoted by white open circles. Times listed below dates represent the time of collection of the first and last sample of the survey (local time).

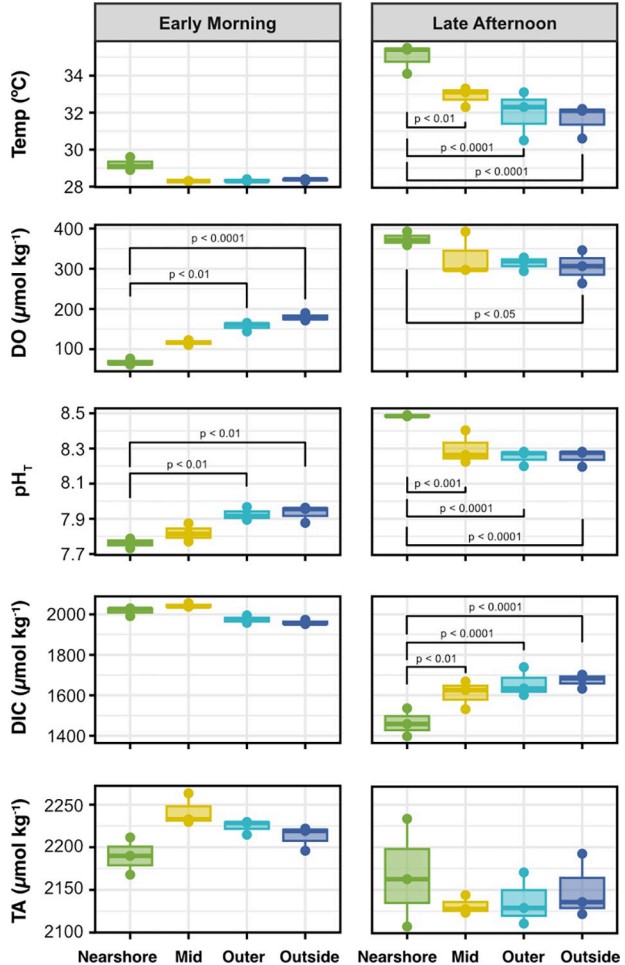

**Fig 3. Spatial gradients in seawater carbonate chemistry and oxygen in the early morning and late afternoon.**
Boxplots of temperature (°C), dissolved oxygen (DO; μmol kg$^{-1}$), total scale pH (pH$_T$), dissolved inorganic carbon
(DIC; μmol kg$^{-1}$); and total alkalinity (TA; μmol kg$^{-1}$) by spatial survey location (nearshore, mid, outer, and outside;
n = 3 each) for the early morning and late afternoon surveys (left and right column, respectively). Significant
differences between pairs of locations (ANOVA and Tukey post-hoc test) are denoted by brackets with listed p-values.

## Metabolic controls of variability

A linear regression of salinity normalized TA (nTA) and DIC (nDIC) from all surveys revealed
a significant, positive relationship (slope = 0.18, R$^2$ = 0.45, p < 0.001) (Fig 4A, 4B and S2
Table) with individual surveys also showing a positive relationship (slopes ranging from 0.15
to 0.46), though this relationship was only significant (p < 0.05) for the mid-morning and
mid-day surveys (S2 Table). Linear regression of dissolved oxygen (DO) and salinity normal-
ized NCP$_{DIC}$ (nDIC$_{NCP}$) from all surveys revealed a significant negative slope (slope = -0.51,
R$^2$ = 0.88, p < 0.001) (Fig 4C, 4D and S2 Table). Linear regressions of DO and nDIC$_{NCP}$ from
individual surveys also revealed significant, negative relationships (slopes from -1.0 to -0.26,
p < 0.05; S2 Table). Observations of both nDIC and nTA and DO and nDIC$_{NCP}$ clustered
strongly by survey, station locations, and pH (Fig 4 and S2 Table).

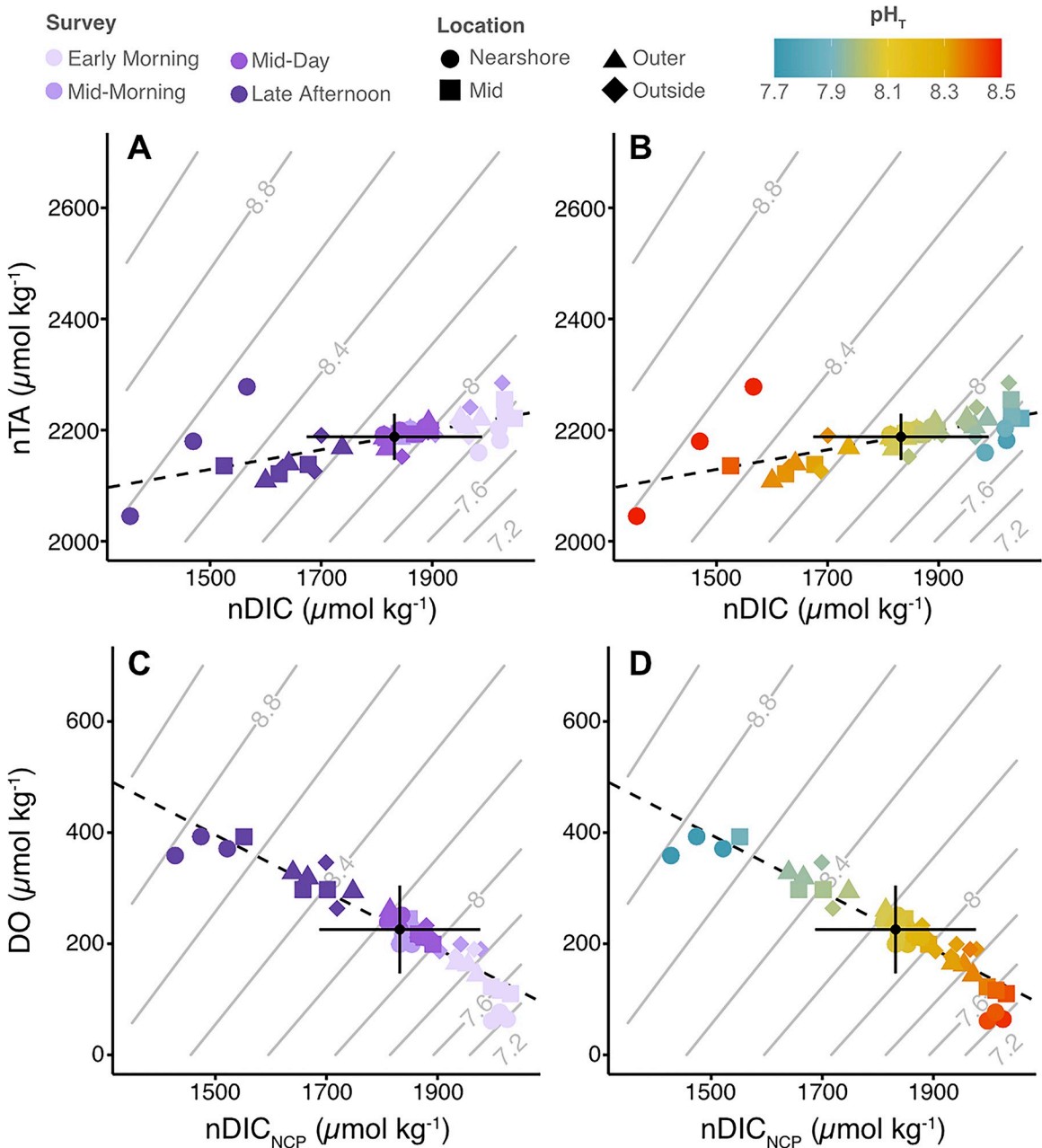

**Fig 4. Salinity normalized TA, DIC, and DO property-property plots.** Salinity normalized total alkalinity (nTA) and salinity normalized DIC (nDIC) shaped by location, colored by: (A) survey time and (B) calculated total scale pH (pH$_T$). Dissolved oxygen (DO) and salinity normalized DIC accounting for NCP effects only (nDIC$_{NCP}$) shaped by location, colored by: (C) survey time and (D) calculated pH$_T$. All normalization was done to a mean salinity of 34.2. Contours in grey represent calculated pH for given nDIC and nTA values. Regression line statistics are reported in S2 Table.

### Spatiotemporal co-variability of temperature, pH, and dissolved oxygen

High-frequency measurements from the CTD sensor deployed at the shallow seagrass site revealed a strong diel pattern of variability in temperature, DO, and pH (S2 Fig). Property-property plots of pH and DO for all spatial survey data (S2A and S2B Fig) and autonomous sensor data (S2C–S2F Fig) revealed a strong, significant relationship between pH and DO

spatially and temporally in the seagrass bed that varied with temperature. Across the 7-day deployment, the temperature varied between 29.08°C to 32.24°C, DO from 0 to 224 μmol kg$^{-1}$ (0 to 121% saturation), and pH from 7.67 to 8.36 (S2C–S2F Fig and S3 Table). Salinity was relatively invariable during dry conditions but decreased in association with precipitation events (33.67 ± 1.08 PSU; mean ± SD) (S2D Fig). Temperature, DO, and pH peaked in the late afternoon, and were lowest in the early morning (S2C, S2E and S2F Fig). Diel variability in temperature (mean daily range ± 1 SD) was 2.28 ± 0.22°C. DO and pH had mean daily ranges of 187 ± 14 μmol kg$^{-1}$ (100 ± 7% saturation) and 0.53 ± 0.02, respectively (S1 Table). The DO concentrations in the seagrass bed reached severely hypoxic concentrations (< 61 μmol $O_2$ kg$^{-1}$ or 2 mg L$^{-1}$; [35]) just before sunrise at the end of each night of the deployment (S2E Fig).

### Coral core linear extension, density, and calcification rates

Annual linear extension rate, density, and calcification rate varied both across years in a single core as well as between cores within and across collection locations (Fig 5 and S3 Fig). Across all cores, years, and collection sites, the mean annual extension rate was 0.94 ± 0.36 cm year$^{-1}$ (mean ± SD), the mean annual density was 2.11 ± 0.40 g cm$^{-3}$, and the mean annual calcification rate was 1.98 ± 0.82 g cm$^{-2}$ year$^{-1}$. We observed no clear temporal trends in extension, density, or calcification rates over the growth record for any site (S3 Fig).

Due to variations in growth rates, core geometry, core length, and banding patterns, the number of usable growth years varied between cores. The core with the longest record had growth years dating to as early as 1994. All cores (N = 15, with poor quality cores removed) contained data for growth years 2012 to 2017. Thus, only data from this 6-year period were used in statistical comparisons between sites. From 2012 to 2017, the mean annual extension

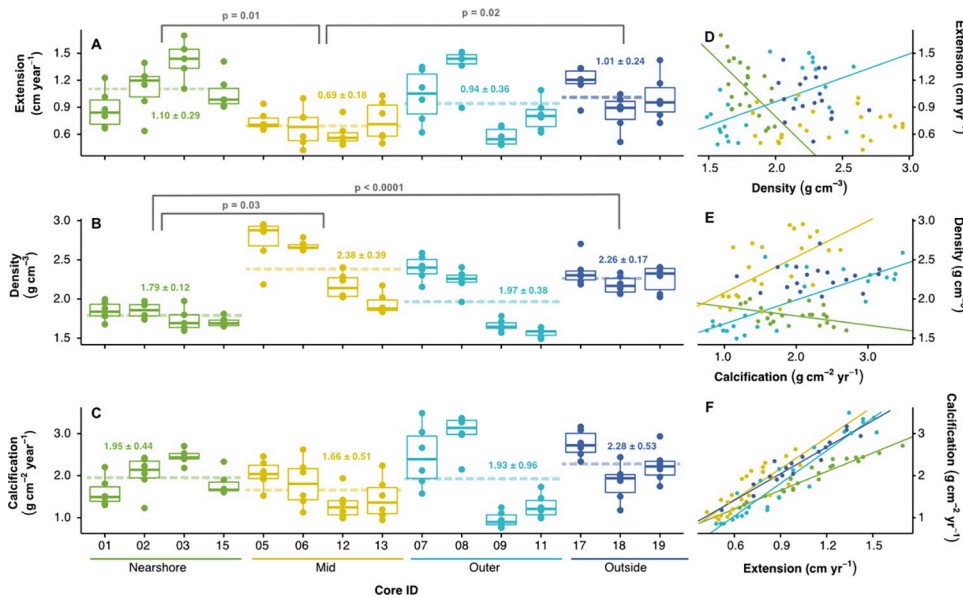

**Fig 5. Differences in coral growth parameters between sites.** (A) Boxplots of annual extension rate (cm year$^{-1}$), (B) density (g cm$^{-3}$), and (C) calcification rate (g cm$^{-2}$ year$^{-1}$) for the years 2012 to 2017 for all cores at each collection location with the mean for the location indicated by a dashed line and the floating value colored by location. Significant differences between mean growth parameters grouped by location are indicated by a grey bracket and corresponding p-value (Tukey; also reported in S4 Table). (D-F) Property-property plots of extension rate, density, and calcification rate for all cores colored by location, with significant linear regression lines colored by location (see S5 Table for detailed statistics).

rate was 0.93 ± 0.31 cm year$^{-1}$ (mean ± SD), the mean annual density was 2.09 ± 0.38 g cm$^{-3}$, and the mean annual calcification rate was 1.93 ± 0.68 g cm$^{-2}$ year$^{-1}$ (S4 Table).

Between 2012 and 2017, linear mixed effects models and a Tukey's pairwise comparison revealed no significant differences in calcification rates between *Porites* from different sites (Fig 5 and S4 Table). However, there were significant differences between annual density and extension rates between *Porites* collected at different sites (Fig 5 and S4 Table). Corals collected from the mid site had significantly higher mean annual density and significantly lower mean annual extension rates compared to corals collected at the nearshore site (p = 0.03 and p = 0.01, respectively; Fig 5 and S4 Table). In addition, the corals collected at the outside site had significantly higher mean annual density than the corals collected at the nearshore site (p < 0.0001) and significantly higher mean annual extension rates compared to the corals from the mid site (p = 0.02; Fig 5 and S4 Table).

Linear regressions of pairs of the growth parameters across all cores from 2012 to 2017 separated by location revealed significant relationships between annual density and extension rate for only the corals from the nearshore and outer sites and in opposing directions (significant and negative slope, or significant and positive slope for *Porites* from the nearshore and outer sites, respectively; Fig 5D and S5 Table). The relationship between annual calcification rate and annual density was significant and positive for the corals from the mid and outer sites, significant and negative for the corals from the nearshore site, and non-significant for the corals at the outside site (Fig 5E and S5 Table). The relationship between annual calcification rate and annual extension rate, however, was strong and significantly positively for all locations (Fig 5F and S5 Table).

## Discussion

### Spatiotemporal variability of carbonate chemistry and dissolved oxygen

Here, we reveal significant gradients in seawater temperature, dissolved oxygen (DO), and carbonate chemistry across the Dongsha seagrass bed from inshore to offshore during our study in the summer of 2018 that have not been previously captured. The spatial gradients were present to some extent during all surveys but were strongest in the early morning and late afternoon (Figs 2 and 3), highlighting the strong nighttime respiration signal and high daytime productivity of the Dongsha seagrass system (Fig 4). The maximum ranges of pH (0.76) and DO (332 μmol O$_2$ kg$^{-1}$) recorded across the Dongsha seagrass bed over all spatial surveys were generally larger than ranges reported from other spatial surveys of tropical and subtropical seagrass beds [24]. Previous work around Dongsha Island has shown that seagrass productivity here is higher than the global average and higher than measurements for the same species in other areas [48, 63]. Thus, these attributes make the Dongsha seagrass site an ideal place to test the influence of natural chemical gradients on coral growth in situ.

Large diel changes were captured by the autonomous sensor deployed in the shallow seagrass, which recorded maximum diel ranges in temperature, DO, and pH of 2.6°C, 204 μmol O$_2$ kg$^{-1}$, and 0.55 pH units. Previous work in seagrass meadows around the globe have reported pH fluctuations ranging from 0.25 to > 1 pH units [12, 19–21, 36, 64, 65] across time scales varying from hours to months. Thus, the 0.55 pH unit range observed in the Dongsha seagrass over just a few days is comparable to what has been measured in other seagrass beds. Notably, DO concentrations as measured by the CTD were severely hypoxic (applying a published threshold of < 61 μmol O$_2$ kg$^{-1}$ or 2 mg L$^{-1}$; [35]) every single morning of the deployment (S2 Fig). This early morning hypoxia was also observed in the spatial surveys, with all of the measurements at the nearshore stations during the early morning survey either below or very close to severely hypoxic (Fig 2). Compared to reef environments dominated mainly by stony corals,

these regular hypoxic conditions are extreme, but are nonetheless projected to increase in duration, frequency, and intensity as the ocean continues to warm and deoxygenate [4].

In contrast to the other parameters measured, total alkalinity was not significantly different between sites during any survey and changes in TA between surveys were relatively small compared to the changes in DIC (Fig 3). Based on the relatively shallow nTA:nDIC slopes of 0.46 and 0.26 from the nTA:nDIC regression analyses for the mid-morning and mid-day surveys (Fig 4A, 4B and S2 Table; [66, 67]), and the stronger relationship between DO and nDIC$_{NCP}$ (slopes of -0.26 to -1.0; Fig 4C, 4D and S2 Table), we can conclude a dominance of organic carbon cycling processes (i.e., photosynthesis and respiration) over calcium carbonate cycling (CaCO$_3$ calcification and dissolution) at this site. This pattern holds across the seagrass bed, and led to the more distinct gradients in DO, pH, and DIC over time compared to TA (Figs 2 and 3). A dominance of organic carbon over CaCO$_3$ cycling has previously been documented in the same seagrass bed from studies spanning both short time scales (i.e., several days) [43] and longer time scales (i.e., multiple seasons over 3 years) [68], with the latter study reporting a comparable nTA:nDIC slope of 0.35 [68]. Similar spatial studies of seagrass beds in Bermuda and San Diego, California reported lower TA:DIC slopes of 0.12 and -0.09, respectively [24]. However, studies of other seagrass beds, such as one conducted at a site in the Red Sea, have reported much higher nTA:nDIC slopes (0.65 ± 0.05; [65]), highlighting that the balance of organic cycling processes versus calcium carbonate cycling varies widely by location.

All of the observed temporal trends in biogeochemical parameters were more pronounced in the shallow, nearshore areas of the seagrass compared to the outside stations, which was likely driven by a combination of shallower depth and higher seagrass density nearshore. The nearshore stations were almost completely covered in seagrass (> 80% cover of seagrass [48]), which led to a higher biomass to water volume ratio compared to the deeper outside stations (which contained mainly sand with some small patches of seagrass). Previous work in other seagrass beds and on coral reefs have demonstrated the importance of depth (and thus the many environmental properties that scale with depth such as light, gas exchange, temperature, etc.) as a property influencing carbonate chemistry variability [69], in addition to flow speed, water residence time, and the trajectory of water flow over the reef [70–72]. Shallower depths allow for the benthic community to modify the above seawater to a greater extent, leading to larger differences along inshore-offshore gradients as well as larger differences between day and night [12, 73].

Importantly, though we observed strong gradients and diel variability in temperature, pH, DIC, and DO during the course of our field study, the short duration of our study limits our conclusions on whether these patterns are maintained seasonally and/or annually. Data from a prior study at the same location revealed seasonal differences in the means and ranges of pH, $p$CO$_2$, DIC, and DO [68]. For example, mean DO saturation was highest in the winter and fall and lowest in the spring and summer [68]. Similarly, mean pH was lowest in the spring and summer compared to the winter and fall [68]. These seasonal variations are likely due to a combination of the effects of temperature and light on seawater chemistry and biological productivity in the seagrass. Other previous work in the same seagrass bed has revealed seasonal and species-specific differences in leaf production rates, with higher production in the summer and fall compared to winter and spring [48], which may drive differences in the strength of these gradients throughout the year. Data collected in other tropical seagrass beds reveal similar seasonal trends [36, 74, 75]. Thus, it is likely that these seasonal changes in mean seawater chemistry ± variability may lead to a dampening of the gradients we observed at different times of year, but additional measurements would be needed to quantify the potential changes in the magnitude of these gradients. However, based on visual assessments of satellite imagery and previous reported coverage [46–48], the seagrass meadow appears persistent over time, suggesting that chemical gradients are also maintained.

## Coral core linear extension, density, and calcification rates

We found that the mean annual extension and calcification rates measured for the corals in the seagrass bed in this study are within values reported for other South China Sea massive *Porites* [76–78] as well as Indo-Pacific massive *Porites* [79–82] living in more traditional reef environments (i.e., coral-dominated, rather than seagrass-dominated). The mean annual extension rate of the seagrass bed *Porites* (0.93 ± 0.31 cm year$^{-1}$) was comparable to linear extension rates from other massive *Porites* cores collected in the South China Sea (0.76 ± 0.37 to 1.27 ± 0.37 cm year$^{-1}$ from [78]; 1.1 cm year$^{-1}$ from [76]) and in other parts of Dongsha Atoll (1.05 cm year$^{-1}$ from [77]). These extension rates are also within the range of reported extension rates for massive *Porites* in the Indo-Pacific (0.54 to 2.67 cm year$^{-1}$; [83]). In contrast, the mean annual density of the seagrass bed *Porites* (2.09 ± 0.38 g cm$^{-3}$) was higher than the density of *Porites* measured in other areas of the Indo-Pacific (0.95 to 1.70 g cm$^{-3}$; [83]).

While data from other Dongsha Atoll coral cores are limited, we observed notable differences in the mean annual extension, density, and calcification rates of the seagrass bed *Porites* compared to *Porites* coral cores collected from a coral-dominated reef site on the eastern reef flat of Dongsha Atoll [84]. The seagrass bed *Porites* from the present study had a lower mean annual extension rate across all growth years (0.93 ± 0.31 cm year$^{-1}$) compared to that of the reef flat *Porites* (1.5 cm year$^{-1}$, no SD provided [84]). However, the mean annual calcification rate for the seagrass bed *Porites* in this study (1.93 ± 0.68 g cm$^{-2}$ year$^{-1}$) was higher than that of the reef flat *Porites* (1.4 ± 0.3 g cm$^{-2}$ year$^{-1}$; [84]). In addition, the mean annual density of the seagrass bed *Porites* from this study (2.09 ± 0.38 g cm$^{-3}$) was also higher than the density measured in the reef flat *Porites* (~ 0.93 g cm$^{-3}$; [84]). While this comparison contrasts with previous studies examining differences in calcification rates between corals residing in similarly variable mangrove lagoons versus reef environments [39], without long-term in situ monitoring it is difficult to identify the drivers of differences in coral growth characteristics between the seagrass bed and reef site with certainty. We note that the reef flat *Porites* were older at collection (70–100 years old) than the corals in the present study (20–25 years old), which may complicate direct comparisons due to the potential age-dependence of skeletal growth rates [86]. However, these two datasets can provide some valuable information on atoll-wide events in Dongsha's past. Notably, in both sets of cores, high density bands were observed during bleaching years–in 2007 and 2015 for reef flat *Porites* [84] and in 2007 for seagrass bed *Porites* in the present study. While we observed no obvious signs of bleaching during 2015 in the seagrass *Porites* cores, we did note high density stress bands in 2007 in 4 of our 15 cores (27%). Similarly, the reef flat study [84] reported stress bands in 6 of 22 (27%) of their cores in the year 2007, suggesting that the 2007 bleaching event was an atoll-wide event and likely impacted massive *Porites* in both environments.

Given the significant seawater temperature, pH, and DO gradients experienced by the corals in the Dongsha seagrass, we expected to see differences in coral growth parameters along these gradients inshore to offshore. Despite high interannual and inter-colony differences in growth parameters, particularly for the *Porites* at the outer site (Fig 5A–5C), we did observe some significant differences in annual extension rate and annual density between the cores from the nearshore site and the cores from the mid and outside sites (Fig 5A–5C). Notably, we found that the cores from both the nearshore and outside sites had significantly higher mean annual extension rates than the cores from the mid site (Fig 5A). It is well-established that linear extension and calcification rates decrease with increasing water depth (and thus decreasing light), while density increases [86]. In the Dongsha seagrass corals, we observed this trend in density where the corals from the nearshore site were significantly less dense than those from the mid and outside sites (Fig 5B). However, the expected trend for extension rates did not

hold across the depth gradient of the collection sites. Other studies of corals along inshore to offshore gradients have recorded variability in coral growth parameters, which were attributed to differences in nutrient availability, heterotrophy versus autotrophy, turbidity, and/or wave action [for a review, see 86]. Thus, we believe that the observed differences in the seagrass bed *Porites* may not be entirely explained by depth or light availability alone and could be due to other environmental parameters we were unable to measure or simply individual differences [83, 85, 86]. In contrast to our original hypothesis, we observed no significant differences in mean annual calcification rates between corals at any site along the seagrass chemical gradient in this study (Fig 5C). While this finding initially contrasts model predictions that coral calcification could be up to 18% higher in the presence of seagrass [12], average *Porites* calcification rates across the seagrass chemical gradient in this study ($1.93 \pm 0.68$ g cm$^{-2}$ year$^{-1}$) were nonetheless higher than that of reef flat corals quantified by [84] ($1.4 \pm 0.3$ g cm$^{-2}$ year$^{-1}$). Based on data from Indo-Pacific massive *Porites* [79, 83, 85], we expected to see a significant inverse relationship between skeletal density and extension rates in the seagrass bed *Porites*. However, we only observed a significant inverse relationship between those parameters for *Porites* growing nearshore (Fig 5D and S5 Table) and found a significant positive relationship for the corals from the outer site (Fig 5D and S5 Table). Similarly, while Indo-Pacific massive *Porites* skeletal density has been reported as either inversely correlated to calcification [79] or not significantly correlated at all [83, 85], we observed a weakly significant positive relationship for the corals at the mid and outer sites, and a significant negative relationship for the nearshore corals (Fig 5E and S5 Table). In line with Indo-Pacific massive *Porites* [79, 83, 85], we observed the strongest relationship between extension and calcification rates for corals at all sites, suggesting that extension rates were the most important drivers of calcification in these corals (Fig 5F and S5 Table), though the nearshore corals had a shallower slope than the corals at the other sites. These contrasting slopes suggest the mechanisms underlying coral skeletal formation across the seagrass bed are likely complex and warrant additional investigation. Regardless, the maintenance of consistent calcification rates across the chemical gradients in the seagrass bed suggest that the corals in the Dongsha seagrass have naturally acclimated to living in these conditions (or were selected for over time) and have thus been able to maintain their calcification rates. Being one of the only coral species to inhabit the seagrass bed suggests that these massive *Porites* may be particularly resistant to environmental stressors [87] compared to other species found on Dongsha Atoll.

## Conclusion

While previous work at this site has explored carbonate chemistry variability in the seagrass bed, the present study is the first to examine the inshore to offshore gradient at fine spatial scales and pair this environmental data with biological data from corals living within these strong gradients. Overall, we demonstrate that seagrass on Dongsha has the ability to strongly modify local water chemistry over both time and space. While the seagrass elevates local pH and DO during the daytime, it also creates acidic and hypoxic conditions at night. Despite these strong gradients, we show that the corals in the Dongsha seagrass bed are persisting and growing at all sampled sites along this extremely dynamic temperature, pH, and dissolved oxygen gradient, though with variations in density and extension rate. While we found no evidence to suggest that corals growing in shallow areas surrounded by seagrass had enhanced calcification rates compared to other corals along the gradient, mean calcification rate and density of the seagrass bed *Porites* was elevated compared to corals previously collected on the reef flat of the atoll. Additional research is needed to investigate whether corals exposed to natural variability in pH and oxygen are better prepared for future exposure to low pH or oxygen.

Future work should continue to leverage seagrass beds as natural laboratories to test how corals will fare under more extreme variable conditions as well as investigate the mechanisms of potential tolerance.

## Supporting information

**S1 Fig. Gradients in temperature, water chemistry, and dissolved oxygen across the seagrass spatial surveys.** Spatial gradients in temperature (˚C), dissolved oxygen (DO; μmol kg$^{-1}$), total scale pH (pH$_T$), dissolved inorganic carbon (DIC; μmol kg$^{-1}$); and total alkalinity (TA; μmol kg$^{-1}$) across the four spatial surveys (stations denoted by black circles) in the Dongsha Island north shore seagrass bed. Coral core collection locations are denoted by gray open circles. Times listed below dates represent the time of collection of the first and last sample of the survey (local time). Delta values reported were calculated as the difference between the survey mean and the value recorded at a given station (i.e., positive values indicate that the value recorded at that station was higher than the survey average and negative values indicate that the value recorded at the station was lower than the survey average).
(TIFF)

**S2 Fig. Spatiotemporal co-variability of temperature, pH, and dissolved oxygen.** Relationship between total scale pH (pH$_T$) and dissolved oxygen (DO; μmol kg$^{-1}$) colored by temperature (˚C) from the (A) spatial seawater surveys and (B) autonomous sensor deployed at a nearshore seagrass site including the slope, R$^2$, and p-value from a linear regression. Time series of (C) temperature (˚C), (D) salinity (PSU), (E) DO (μmol kg$^{-1}$ [left] and percent saturation [right]), and (F) pH$_T$ from the IDRONAUT CTD sensor deployed in the shallow nearshore seagrass. Gray shaded boxes denote local nighttime hours. Dashed line in E denotes severe hypoxia threshold (61 μmol kg$^{-1}$).
(TIFF)

**S3 Fig. Trends of coral skeletal growth parameters over time.** Time series of annual extension rate (cm year$^{-1}$), density (g cm$^{-3}$), and calcification rate (g cm$^{-2}$ year$^{-1}$) for all years and cores at each of the four collection locations (columns; n = 4 cores at nearshore, mid, and outer sites, and n = 3 at the outside site). Lighter color lines are time series for individual cores and the darker line is the mean for all cores at that site with 95% confidence interval shading above and below.
(TIFF)

**S1 Table. Seagrass spatial survey statistics.** Minimum, maximum, range, and mean (± 1 standard deviation; n = 12 per survey) temperature, salinity, dissolved oxygen concentration, dissolved oxygen percent saturation, total scale pH (pH$_T$), dissolved inorganic carbon, and total alkalinity for each spatial survey from in situ YSI measurements and bottle sample data.
(DOCX)

**S2 Table. Linear model statistics for Fig 4.** Slope, intercept, R$^2$, and p-value for regressions between salinity normalized total alkalinity (nTA) and dissolved inorganic carbon (nDIC) as well as dissolved oxygen (DO) and salinity normalized net community productivity derived from DIC (nNCP$_{DIC}$) across all surveys and for individual surveys. Regression lines associated with these statistics for all surveys grouped (first column only) are plotted in Fig 4. Asterisks indicate level of significance (* p ≤ 0.05, ** p ≤ 0.01, *** p ≤ 0.001).
(DOCX)

**S3 Table. Seagrass CTD sensor statistics.** Daily statistics (mean daily minimum, maximum, range, and mean ± 1 SD) and deployment absolute minima and maxima for temperature,

salinity, dissolved oxygen (DO) concentration, dissolved oxygen percent saturation, and total scale pH (pH$_T$) recorded by the autonomous sensor in the shallow seagrass. Only full days (i.e., complete 24-hour cycle) are included in daily means (2–7 July 2018).
(DOCX)

**S4 Table. Mean coral core growth parameters by site between 2012 and 2017 as reported in Fig 5A-5C.** Mean annual extension, density, and calcification rates between 2012 and 2017 for corals at the nearshore (n = 4), mid (n = 4), outer (n = 4), and outside sites (n = 3), as well as averaged across all sites (n = 15). Statistically significant differences between sites for extension, density, and calcification rates are indicated with superscript letters and asterisks indicating level of significance for each column (* $p \leq 0.05$, ** $p \leq 0.01$, *** $p \leq 0.001$).
(DOCX)

**S5 Table. Linear regression statistics for Fig 5D-5F.** Slope, intercept, $R^2$, and p-value for linear regressions of coral growth data from 2012 to 2017 for each site (Nearshore, Mid, Outer, Outside). Regression lines associated with these statistics are plotted in Fig 5D-5F (if significant, as denoted by asterisks: * $p \leq 0.05$, ** $p \leq 0.01$, *** $p \leq 0.001$).
(DOCX)

**S1 File. Inclusivity in global research checklist.** Information regarding ethical considerations of research conducted internationally for this manuscript.
(DOCX)

## Acknowledgments

We thank National Sun Yat-sen University, the Coast Guard Administration of the Republic of China (Taiwan), the Dongsha Atoll Marine National Park, Oscar Ng, Yuli Chiu, Hai-Jin Zhang, Di Bing Tam, Jeanie Lim, and Rong-Wei Syu for their help in coordinating and assisting with fieldwork. We also thank Dr. Phil Hastings and the Scripps Institution of Oceanography Marine Vertebrate Collection for access to x-radiography instruments as well as Dr. Jessica Carilli and Dr. Richard Norris for access to the aragonite and aluminum wedges used in the x-radiography.

## Author Contributions

**Conceptualization:** Ariel K. Pezner, Travis A. Courtney, Andreas J. Andersson.

**Data curation:** Ariel K. Pezner.

**Formal analysis:** Ariel K. Pezner.

**Funding acquisition:** Ariel K. Pezner, Keryea Soong, Andreas J. Andersson.

**Investigation:** Ariel K. Pezner, Travis A. Courtney, Hui-Chuan Chu, Benjamin W. Frable, Samuel A. H. Kekuewa, Yi Wei, Andreas J. Andersson.

**Methodology:** Ariel K. Pezner, Travis A. Courtney, Andreas J. Andersson.

**Project administration:** Keryea Soong, Yi Wei, Andreas J. Andersson.

**Resources:** Wen-Chen Chou, Benjamin W. Frable, Keryea Soong, Yi Wei, Andreas J. Andersson.

**Software:** Ariel K. Pezner.

**Supervision:** Travis A. Courtney, Andreas J. Andersson.

**Visualization:** Ariel K. Pezner.

**Writing – original draft:** Ariel K. Pezner.

**Writing – review & editing:** Ariel K. Pezner, Travis A. Courtney, Wen-Chen Chou, Hui-Chuan Chu, Benjamin W. Frable, Samuel A. H. Kekuewa, Keryea Soong, Yi Wei, Andreas J. Andersson.

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
