## [Decision Letter · Decision Letter 0]

17 Jan 2024

PONE-D-23-37336Coral growth along a natural gradient of seawater temperature, pH, and oxygen in a nearshore seagrass bed on Dongsha Atoll, TaiwanPLOS ONE

Dear Dr. Pezner,

Thank you for submitting your manuscript to PLOS ONE. After careful consideration, we feel that it has merit but does not fully meet PLOS ONE’s publication criteria as it currently stands. Therefore, we invite you to submit a revised version of the manuscript that addresses the points raised during the review process.

We look forward to receiving your revised manuscript.

Kind regards,

Silvia Mazzuca

Academic Editor

PLOS ONE

Journal Requirements:

3. Please include a complete copy of PLOS’ questionnaire on inclusivity in global research in your revised manuscript. Our policy for research in this area aims to improve transparency in the reporting of research performed outside of researchers’ own country or community. The policy applies to researchers who have travelled to a different country to conduct research, research with Indigenous populations or their lands, and research on cultural artefacts. The questionnaire can also be requested at the journal’s discretion for any other submissions, even if these conditions are not met. 

Please find more information on the policy and a link to download a blank copy of the questionnaire here: https://journals.plos.org/plosone/s/best-practices-in-research-reporting. Please upload a completed version of your questionnaire as Supporting Information when you resubmit your manuscript.”

This research was supported by the National Science Foundation (https://www.nsf.gov/) grants OCE-1255042 (AJA) and OCE-1829778 (AJA), a National Science Foundation Graduate Research Fellowship DGE-2038238 (AKP), and Philanthropic Educational Organization (P.E.O.; https://www.peointernational.org/peo-scholar-awards) Scholar Award (AKP). These funders did not play any role in study design, data collection and analysis, decision to publish, or preparation of the manuscript.

5. We noted in your submission details that a portion of your manuscript may have been presented or published elsewhere. The dissolved oxygen and temperature data only from the autonomous sensor in the seagrass was used in a recent publication by Pezner et al. in Nature Climate Change (2023), which compiled a global dataset of dissolved oxygen in reef habitats around the world. As this dataset is presented in full in the current MS (temperature, pH, dissolved oxygen, salinity) and is only a small part of the current analysis, we do not believe this constitutes dual publication. These data are only included in the present manuscript only to illustrate the temporal variability in environmental and chemical parameters that the corals are exposed to, but do not constitute the main findings of this study. Please clarify whether this [conference proceeding or publication] was peer-reviewed and formally published. If this work was previously peer-reviewed and published, in the cover letter please provide the reason that this work does not constitute dual publication and should be included in the current manuscript.

7. We note that Figure 1 in your submission contain map and satellite images which may be copyrighted. All PLOS content is published under the Creative Commons Attribution License (CC BY 4.0), which means that the manuscript, images, and Supporting Information files will be freely available online, and any third party is permitted to access, download, copy, distribute, and use these materials in any way, even commercially, with proper attribution. For these reasons, we cannot publish previously copyrighted maps or satellite images created using proprietary data, such as Google software (Google Maps, Street View, and Earth). For more information, see our copyright guidelines: http://journals.plos.org/plosone/s/licenses-and-copyright.

We require you to either present written permission from the copyright holder to publish these figures specifically under the CC BY 4.0 license, or remove the figures from your submission:

Additional Editor Comments:

Dear Author,

both the reviewer and I have evaluated your manuscript. The manuscript is comprehensible. However, as indicated in reviewer comments , I advise considering the publication of a significantly more concise version. Several figures and tables could be moved to the supplement, and the text, particularly in the discussion section, can be substantially reduced (the reviewer has provided some suggestions, but an overall rigorous edit is necessary). The primary value lies in the data, which merit publication. Nevertheless, to align with the standards of PLOS One, the paper should be concise, with the majority of information presented in the supplement.

after this deep revision the manuscript will deserve publication

best egrads

Silvia Mazzuca

Handling Editor

Reviewers' comments:

Reviewer's Responses to Questions

**Comments to the Author**

1. Is the manuscript technically sound, and do the data support the conclusions?

Reviewer #1: Yes

2. Has the statistical analysis been performed appropriately and rigorously? 

Reviewer #1: Yes

3. Have the authors made all data underlying the findings in their manuscript fully available?

Reviewer #1: No

4. Is the manuscript presented in an intelligible fashion and written in standard English?

Reviewer #1: Yes

5. Review Comments to the Author

Reviewer #1: Review- Manuscript Number: PONE-D-23-37336

‘Coral growth along a natural gradient of seawater temperature, pH, and oxygen in a nearshore seagrass bed on Dongsha Atoll, Taiwan’

The manuscript describes a well-executed and technically sound study that describes spatial patterns in water chemistry (DO, pH, DIC, TA, Temp) across an inshore-offshore gradient around Dongsha Island. The study site is an interesting habitat with a mixed seagrass and coral community and well-suited to test the hypothesis that primary producers convey an “ameliorating effect” to coral performance under increased CO2 availability. However, the study was only over a very short period of time, a week in June 2018. This limits its value, especially with regard to being able to interpret the data from the Porites cores as the main indicators of coral performance at this site.

The manuscript clearly complements earlier work, which conducted more measurements of water chemistry in the seagrass beds at the study site, Dongsha Island [eg ref 55, study over a year in 2011] and analysed coral cores in a neighbouring non-seagrass habitat [eg ref 58, collected in 2015, analysed 2007-2012]. To increase the value of the MS, I suggest that the authors more clearly outline how the current study builds on and/or complements this earlier work, and perhaps even consider presenting some of the earlier data in tables or analyses.

As it stands, the MS is descriptive and its value is in the data (which I understand will be publicly available at publication), which others may use for future studies or metanalyses. For this, the MS is publishable (after revision) - but my recommendation is to publish a much cut-back version (see some suggestions below in my detailed comments).

In my opinion, it is likely that the relatively young Porites colonies studied here are tolerant (eg selection during recruitment) to the local, diurnally highly variable, seawater chemistry (which is driven by the seagrass metabolism at 3 of the 4 sites). Hence, it is not surprising that the differences in Porites calcification, extension and density were less pronounced than expected by the authors. As the authors correctly conclude, other variables than the measured might have influenced the observed patterns. Also, interactions between the various environmental variables are not well understood and could also have an influence.

The discussion occasionally strays into speculation, and I recommend the authors take care not to over-interpret the limited available environmental data.

As an aside: The coral core data analysis was limited to only the years 2012-2017 (for QA reasons)- this for example does not include periods of potential early stages of recovery from the 2007 bleaching event which was record in (most?) analysed cores. This would have been interesting as the water chemistry at the 4 locations might have modulated the recovery trajectory.

Detailed comments:

Introduction:

Note that correct citation for ref [5]is: Souter, D., Planes, S., Wicquart, J., Logan, M., Obura, D., Staub, F. (eds) (2021). Status of coral reefs of the world: 2020 report. Global Coral Reef Monitoring Network (GCRMN) and International Coral Reef Initiative (ICRI). DOI: 10.59387/WOTJ9184.

Ref [6] is quite dated, lots of more recent refs after the 2015-17 bleaching

Methods:

Describe somewhere how abundant the corals (esp Porites) are at your sites.

Line 206: write out NCP when first used here

Depths reported should be normalised to an appropriate tide datum (eg line 240/241), eg LAT

Line :253 reword, sloppy language” Cores were slabbed”

Results:

Line 301: suggest that Table 1 is moved to supplement

Line 344/345: not so surprising, this is just a point in time and v localised factors could be at play here…

Line 403: hypoxia every end of the night/before sunrise, not “evening”

Line 405: Fig 5- I would recommend gray shading for nighttime hours to make the graph more intuitive…

I find in Fig 5 the correlation between DO and pH and colour coded by temperature a bit misleading as it is perhaps more time of day and photosynthesis influencing this, as clearly illustrated in panel 5 c to f…

Line 413: suggest that Table 1 is moved to supplement

Line 421 and para below: Fig 6- what is reason for v high variability in parts of record?

Line 449: omit referring to Table 3 when describing the statistical comparison between cores, this information is not in Table 3 (but should be). For this section I suggest that Fig 7 (d - f) Table 3 contains sufficient information (if statistical results are added as eg asterisks) to support the results and Fig 6 and Fig 7 a-c could be included in the supplementary material w/o losing much content.

Discussion:

Line 497: the SSP predictions are global averages- comparing the locally measured dial ranges of values to the predictions is not meaningful.

Line 504: suggestion for rewording “(applying published thresholds of < 61 µmol O2 kg-1or 2 mg L-1504 ; [35])”

Line 507: “Compared to traditional reef environments, these regular hypoxic conditions are unique, but are nonetheless projected to increase in duration, frequency, and intensity as the oceans continue to warm and deoxygenate [4].” - what do you consider “traditional reef environments”, suggest something like: ‘’coral reef communities without substantial seagrass biomass’. Ref [4] is predictions for coral reef environments.

Line 519 onwards: Some of the referenced studies appear to be from just seagrass beds not a mixed community such as the one studied here. Are there indications that the primary producers dominate the water chemistry in mixed communities and the fact that some corals are present is not expected to make much of a measurable difference (hence comparisons to seagrass-only habitats is valid)?

Para 526-543: discussion here overly convoluted, the points here could be made more succinctly.

Line 527: suggest “more pronounced” instead of “heightened”

Lines 528- 530: normalise to tide reference

Lines 530-532: “almost complethey covered”- as you have no data on the seagrass biomass I suggest you are bit more descriptive here- close to 100% cover of dense seagrass vs some sort of description of what the ‘outside’ seagrass community looked like, in comparison.

Line 535: depth as a property is in most cases a proxy for variety of env variables that change along deoth gradients: eg benthic light, temperature, volume of overlying water, importance of gas exhancge at surface etc.

Line 537: Not sure what you mean by “path of the water over the reef”- you mentioned the relevant variables, flow speed, residence time (which are also related)

Line 538: “modify the above seawater”- unusual expression, this is just physics, ie area of seafloor vs overlying water volume, and diffusion/dispersion of metabolites…

Line 574 (and elsewhere): “seagrass corals”- sloppy term…

Line 598-600: this is speculation

Line 611: suggest to add “meadow” or “bed” to ‘seagrass’ and saying Porites instead of ‘corals’

Line 613/614: suggest: “Porites at the outer sites” and “cores from the mid and outer sites”

Line 660 & Line 680: Porites appears to be particularly tolerant to a many env variables, not just low pH/high DIC

Line 682: Consider adding this reference, showing that a previously reported decline in calcification in GBR Porites was only transitory:

Cantin NE, Lough JM (2014) Surviving Coral Bleaching Events: Porites Growth Anomalies on the Great Barrier Reef. PLoS ONE 9: e88720 doi 10.1371/journal.pone.0088720

Line 692: delete “and dissolved oxygen concentrations”, next sentence explains the changes in water chemistry you measured.

6. PLOS authors have the option to publish the peer review history of their article (what does this mean?). If published, this will include your full peer review and any attached files.

Reviewer #1: No

---

## [Author Response · Author response to Decision Letter 0]

8 Apr 2024

We thank the Editor and Reviewer for their helpful suggestions and comments. Please see the "Response to Reviewers" document for detailed responses to each comment.

---

## [Decision Letter · Decision Letter 1]

4 Oct 2024

Coral growth along a natural gradient of seawater temperature, pH, and oxygen in a nearshore seagrass bed on Dongsha Atoll, Taiwan

PONE-D-23-37336R1

Dear Dr. Pezner,

We’re pleased to inform you that your manuscript has been judged scientifically suitable for publication and will be formally accepted for publication once it meets all outstanding technical requirements.

Kind regards,

Goulven G Laruelle

Academic Editor

PLOS ONE

Additional Editor Comments (optional):

Dear authors, I apologize for letting this answer drag for such a long time. Several reviewers accepted to evaluate your resubmission but never submitted their report. One of the previous reviewers however did evaluate your answers and updated manuscript, considering that it should now be accepted for publication. Based on my own reading and evaluation of your resubmission, I also agree with this conclusion and do not want to let you wait any longer. Please note the remaining remark of the reviewer that could take into account when sending your files for production.

Reviewers' comments:

Reviewer's Responses to Questions

**Comments to the Author**

1. If the authors have adequately addressed your comments raised in a previous round of review and you feel that this manuscript is now acceptable for publication, you may indicate that here to bypass the “Comments to the Author” section, enter your conflict of interest statement in the “Confidential to Editor” section, and submit your "Accept" recommendation.

Reviewer #1: All comments have been addressed

2. Is the manuscript technically sound, and do the data support the conclusions?

Reviewer #1: Yes

3. Has the statistical analysis been performed appropriately and rigorously? 

Reviewer #1: Yes

4. Have the authors made all data underlying the findings in their manuscript fully available?

Reviewer #1: Yes

5. Is the manuscript presented in an intelligible fashion and written in standard English?

Reviewer #1: Yes

6. Review Comments to the Author

Reviewer #1: The only additional suggestion I would like to make is to add another reference to Line 568-570:

"Being one of the only coral species to inhabit the seagrass bed suggests that these massive Porites may be particularly resistant to environmental stressors compared to other species found on Dongsha Atoll."

A traits analysis in https://doi.org/10.1111/j.1461-0248.2012.01861.x suggests that massive Porites species are stress-tolerant. This is also something found in many papers looking at various environmental stressors. Noteworthy in the OA context is also DOI: http://dx.doi.org/10.1038/nclimate1122 (which was Ref 99 but was cut in the revision). I don't feel strongly about this but think it is informative to know that the corals that live in the mixed seagrass community at the study site are likely to be generally tolerant anyway...

7. PLOS authors have the option to publish the peer review history of their article (what does this mean?). If published, this will include your full peer review and any attached files.

Reviewer #1: No

---

## [Editor Report · Acceptance letter]

11 Oct 2024

PONE-D-23-37336R1 

PLOS ONE

Dear Dr. Pezner, 

I'm pleased to inform you that your manuscript has been deemed suitable for publication in PLOS ONE. Congratulations! Your manuscript is now being handed over to our production team.

Kind regards, 

on behalf of

Dr. Goulven G Laruelle 

Academic Editor

PLOS ONE